# Real-space observation of incommensurate spin density wave and coexisting charge density wave on Cr (001) surface

Yining Hu [1,5], Tianzhen Zhang[1,5], Dongming Zhao[1,5], Chen Chen[1], Shuyue Ding[1], Wentao Yang[1], Xu Wang[1], Chihao Li[1], Haitao Wang[1], Donglai Feng [2,3,4] & Tong Zhang [1,3,4✉]

In itinerant magnetic systems, a spin density wave (SDW) state can be induced by Fermi surface nesting and electron-electron interaction. It may intertwine with other orders such as charge density wave (CDW), while their relation is still yet to be understood. Here via spin-polarized scanning tunneling microscopy, we directly observed long-range spin modulation on Cr(001) surface, which corresponds to the well-known incommensurate SDW of bulk Cr. It displays 6.0 nm in-plane period and anti-phase behavior between adjacent (001) planes. Meanwhile, we simultaneously observed the coexisting CDW with half the period of SDW. Such SDW/CDW have highly correlated domain structures and are in-phase. Surprisingly, the CDW displays a contrast inversion around a density-of-states dip at −22 meV, indicating an anomalous CDW gap opened below $E_F$. These observations support that the CDW is a secondary order driven by SDW. Our work is not only a real-space characterization of incommensurate SDW, but also provides insights on how SDW and CDW coexist.

[1] State Key Laboratory of Surface Physics, Department of Physics, and Advanced Materials Laboratory, Fudan University, 200438 Shanghai, China. [2] Hefei National Laboratory for Physical Science at Microscale and Department of Physics, University of Science and Technology of China, 230026 Hefei, Anhui, China. [3] Collaborative Innovation Center of Advanced Microstructures, 210093 Nanjing, China. [4] Shanghai Research Center for Quantum Sciences, 201315 Shanghai, China. [5] These authors contributed equally: Yining Hu, Tianzhen Zhang, Dongming Zhao. ✉email: tzhang18@fudan.edu.cn

A spin density wave (SDW) state manifests itself as real-space spin modulations. It is usually formed in itinerant magnetic systems with Fermi surface nesting and electron–electron interactions[1]. The spatial period of SDW could be commensurate (C-SDW) or incommensurate (IC-SDW) to lattice constant. In the latter case, the spin modulation decouples from lattice, which is distinguished from local moment induced anti-ferromagnetic (AFM) order. Interestingly, SDW often coexists and sometimes intertwines with other orders in correlated systems, such as charge density wave (CDW) and superconductivity[1–4]. The interplay of these coexisting/intertwining orders has now become an important theme in condensed matter physics. To date, the commonly observed SDW states are commensurate SDW, such as the collinear/bicollinear SDW (AFM) state in iron-based superconductors[4,5]; while incommensurate SDW is rarely seen, and particularly, its real space imaging is quite lacking.

Chromium (Cr) is one of the classic examples which shows itinerant magnetism with an IC-SDW ground state[6–8] below its Néel temperature ($T_N = 311$ K). Such IC-SDW is stabilized by "imperfect" Fermi surface nesting condition[9], as illustrated in Fig. 1a. Specifically, the Fermi surfaces of Cr (b.c.c. lattice) are composed of hole pockets at the corner and electron pocket at the center of the Brillouin zone[6]. The hole pocket is slightly larger than the electron pocket which yields two nesting vectors: $Q_{\pm} = 2\pi/a\ (1 \pm \delta)$ ($a = 2.9$ Å being the lattice constant). Therefore, a long period IC-SDW with a wave vector $Q_{SDW} = 2\pi\delta/a$ is generated which overlaps with the AFM coupling between Cr atoms (Fig. 1b). The wavelength of IC-SDW is reported to be 6.0 nm at $T < 10$ K (ref. [10]), and $Q_{SDW}$ is along one of the <001> directions. The spin orientation of Cr atom is found to be perpendicular to $Q_{SDW}$ at $T > T_{SF}$ (123 K) but switched to be parallel at $T < T_{SF}$ (spin-flip transition[6]).

In addition to IC-SDW, a charge density wave (CDW) with half period of the IC-SDW was also found in Cr[10–13]. Unlike the IC-SDW, the exact origin of such CDW is yet to be understood. It was often considered as the second-order harmonics of IC-SDW[12], corresponding to a nesting vector $Q_{CDW} = 2Q_{SDW}$ that connects the two folded bands at Γ (Fig. 1a); alternatively, it was suggested as a lattice strain wave induced by magneto-elastic coupling to the IC-SDW[6]. Therefore, being a pure element arranged in a simple structure, Cr is also a classical system to study the interplay of SDW/CDW orders.

However, after decades of research, the characterization of IC-SDW (and CDW) in Cr is still rather limited to spatially averaged method, such as neutron scattering[6,7], x-ray diffraction[10,13] and photoemission spectroscopy[14,15]. In principle, SDW could also be detected by local probes at atomic scale, such as spin-polarized scanning tunneling microscopy (SP-STM)[16]. Although a few SP-STM studies have been performed on various Cr surfaces[17–26], the real-space evidence of IC-SDW was rarely reported (some studies found CDW modulation on Cr (110) surface[23,27], and argued the satellite FFT spots as an indication of IC-SDW[23]). Most SP-STM studies on Cr (001) surface only observed in-plane ferromagnetism with AFM coupling between adjacent (001) planes[17–22]. To understand such a ferromagnetic spin arrangement on surface, it was suggested the magnetic moment is enhanced at the surface[28] and the IC-SDW antinodes are always pinned on the surface[8,19], making it invisible to STM. However, we noticed that most previous STM studies on Cr (001) did not resolve clear atomic lattices, although mono-atomic terrace can be identified. This is likely due to local disorders induced by segregated impurities on the surface, which is a common problem in cleaning Cr single crystal. As the surface conditions could alter the surface magnetism[8], it would be intriguing to search the IC-SDW in real space again on a well-ordered Cr surface.

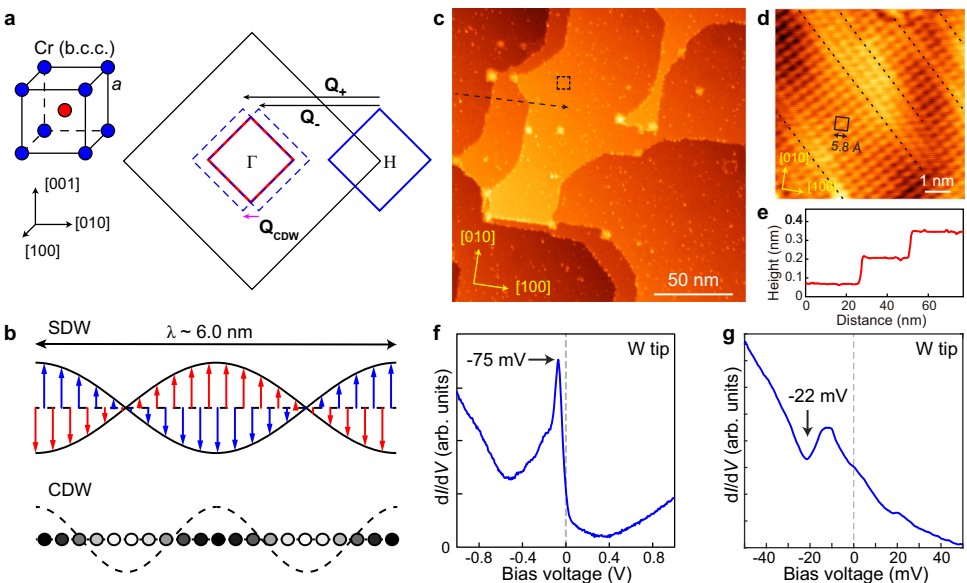

**Fig. 1 Sketch of the crystal structure and SDW/CDW states of bulk Cr, and STM characterization of Cr(001) surface. a** Left: the body-centered cubic (b.c.c.) structure, and right: the (001) plane Brillouin zone of Cr. Electron and hole Fermi surfaces (cross sections) are represented by red and blue squares, respectively. The nesting vectors $Q_{\pm} = 2\pi/a\ (1 \pm \delta)$ are indicated by black arrows, and $Q_{CDW} = Q_{+} - Q_{-}$ is the CDW wave vector. **b** Schematic of incommensurate SDW and CDW of bulk Cr in real space. Blue and red arrows represent the spin of corner and body-center Cr atoms in the b.c.c. lattice at $T > T_{SF}$, respectively. Solid (hollow) circles represent the locations with the highest (lowest) charge density. **c** Large scale STM image (190 × 190 nm²) of the cleaned Cr (001) surface. **d** Atomically resolved STM image (taken in the dashed square in panel **c**) showing a centered (2 × 2) structure. The dashed lines indicate the surface line-dislocations (see Supplementary Fig. S1 for more details). **e** Line profile taken along the dashed line in panel (**c**). **f** Typical d$I$/d$V$ point spectrum taken on a defect-free region with a normal W tip (setpoint: $V_b = 1$ V, $I = 60$ pA, $\Delta V = 20$ mV). **g** Typical d$I$/d$V$ spectrum taken around $E_F$ by W tip (setpoint: $V_b = 50$ mV, $I = 100$ pA, $\Delta V = 1$ mV), a DOS dip at $V_b \approx -22$ (±1) mV is observed.

In this work, by using low-temperature spin-polarized STM with vector magnetic field, we studied a thoroughly cleaned Cr (001) with a well-ordered surface. We observed clear spin modulation with a period of 6.0 nm, propagating along in-plane [100] or [010] directions, which well matches the projected bulk IC-SDW on (001) surface. Its SDW nature is confirmed by the contrast inversion upon switching tip's magnetization, and the anti-phase relation between adjacent terraces. Meanwhile, we also observed the coexisting CDW with a period 3.0 nm, and surprisingly found that it displays a $\pi$ phase shift around gap structure about 22 meV below $E_F$, which suggests its formation is beyond the intuitive Fermi surface nesting picture. Furthermore, as a local probe measurement, we directly observed the domain structure of SDW/CDW and revealed their in-phase relation. Our work not only gives a real-space investigation of IC-SDW, but also provide new insights on the general mechanism of coexisting SDW/CDW orders.

## Results

### STM characterization of Cr (001) surface and it's tunneling spectrum

The experiment was conducted in a cryogenic STM (UNISOKU) at $T = 5.0$ K. Details about the cleaning process of Cr (001) and STM measurement are described in the "Methods" section. Figure 1c shows a large scale STM image of the obtained Cr (001) surface. It displays atomically flat terraces with mono-atomic height ($\approx 0.14$ nm), as indicated by the line profile in Fig. 1e. It is notable that the terrace edges prefer running along high symmetric directions such as [100], [010], and [110]. This is an indication of free surface atom diffusion during annealing[29]. Despite some randomly distributed defects, atomic lattice can be easily resolved in defect free area, as shown in Fig. 1d. It displays a centered $2 \times 2$ (or $\sqrt{2} \times \sqrt{2}$ R45°) lattice with respect to the pristine Cr $1 \times 1$ lattice. Some dislocation lines are observed where the atomic lattice on their two sides displays certain shift (black dashed lines in Fig. 1d). These dislocations do not show influence to SDW/CDW discussed below, more details are presented in Supplementary Fig. S1. We noticed some previous STM works on Cr (001) also observed c $(2 \times 2)$ structures[22,26,30], but the electronic states of the present surface is quite different from those studies (shown below). Although the origin of this reconstruction is unclear at this stage, it is the first time to observe regular oriented terrace edges with well-ordered lattice on a sputtered/annealed Cr (001) surface.

The typical large energy scale dI/dV spectrum of the surface, measured by a normal W tip above defect-free area, is shown in Fig. 1f. There is a pronounced DOS peak located at $-75$ ($\pm 5$) mV. We note although a DOS peak was widely observed on Cr surfaces[18–26,31,32], the peak position varies significantly for different studies. The origin of such a peak was usually attributed to spin-polarized surface state[31,33–35] or the orbital Kondo effect[32]. Our measurements shown below tend to support the former scenario. By zooming into a narrower energy range near $E_F$ (Fig. 1g), we found there is an additional DOS dip at $E = -22 (\pm 1)$ meV, which has never been reported before. Such a DOS dip is repeatedly observed at different surface locations (see Supplementary Fig. S2 for more spectra). We will show later that it is likely an energy gap associated with the CDW order, but opens below the Fermi level.

### Real-space imaging of incommensurate SDW

We then studied the surface with spin-polarized tips. Fig. 2a is a dI/dV map taken at $V_b = -150$ meV with a tip coated with 40 nm thick Cr, which favors an in-plane spin polarization[16]. The mapping area is the same as that shown Fig. 1c. It is remarkable that stripe-like modulations can be observed, and there are two domains of such

modulation which are perpendicular to each other. A zoomed-in dI/dV map around a domain wall is shown in Fig. 2b. The period of the stripe is 6.0 nm and the wave vector is either along [010] or [100] direction (as also seen in the FFT image in Fig. 2a inset). Such a period and propagating direction exactly match the projected bulk IC-SDW of Cr on a (001) surface. It is also seen that the domain walls in Fig. 2a (dashed curve) have no correlation with surface morphology (Fig. 1c), which indicates the stripes are not merely surface effects but of bulk origin. Figure 2c shows the typical dI/dV spectra taken on the stripe and between the stripes with the Cr-coated tip. There is observable difference on the intensity around the DOS peak, which is attributed to spin contrast as discussed below. More dI/dV maps taken at different energies and their FFT images can be found in Supplementary Figs. S3 and S4.

To further verify these stripes are spin modulations, we performed measurement with a 16 nm-Fe-coated tip whose magnetization can be controlled by an external magnetic field[16]. Figure 3a, b are two dI/dV maps taken in the same region, but under opposite in-plane field of $B_X = \pm 1$ T (**X** direction is perpendicular to stripes, as marked in figure). The 6.0 nm period stripes can be seen in both Fig. 3a, b, while they display a clear phase inversion, as further illustrated in their line profiles in Fig. 3e. Since the tip magnetization will follow such in-plane field, this gives a direct evidence that the stripes are SDW modulations, with opposite spin orientations on their peaks and troughs. We can also tune the tip magnetization along **Y** and **Z** directions (by applying $B_Y = 1$ T and $B_Z = 1.5$ T, respectively), the resulting dI/dV maps are shown in Fig. 3c, d, respectively. In these two cases, the contrast of the stripes was significantly reduced and almost invisible. This confirmed that the spins are only polarized along **X** direction (the same direction of $\mathbf{Q}_{SDW}$), which agrees with bulk measurements that the IC-SDW is longitudinal wave at $T < T_{SF}$ (ref. [6]). We can extract the spin-polarization ratio (spin-contrast) of Fig. 3a, b by calculating their relative intensity difference, which is about 4% at the SDW peaks (Fig. 3f). We note here that for Cr-coated tip used in Fig. 2, its (in-plane) polarization direction is arbitrary[16], that is why the two SDW domains in Fig. 2a, b can be simultaneously imaged but their contrasts are different.

We note that the above measurement under vector magnetic field also distinguished the SDW state from "spin-spiral" order which has been detected by SP-STM in other magnetic systems[36,37]. In a spin-spiral, the spins have a nearly constant magnitude but their orientations keep rotating with certain chirality, thus one would observe spin modulation in at least two of the $X$, $Y$, $Z$ components (depends on the types of spin-spiral, e.g., helical or cycloidal[36]). However here we only observed spin modulation along X direction.

Another signature of IC-SDW can be obtained near the atomic step edges. Figure 3g shows a topographic image of three adjacent mono-atomic height terraces, and Fig. 3h is the corresponding dI/dV map measured by a Cr-coated tip. As shown by dashed lines, there is a phase inversion of the modulations between adjacent terraces. This indicates the local spin of two adjacent (001) plane are still AFM coupled. Based on above observations, we now achieve a complete spin configuration of the present Cr (001) surface. As illustrated in Fig. 3i, the Cr have an out-of-plane AFM configuration with the spins lying in-plane, while a long wavelength, longitudinal IC-SDW ($\lambda = 6.0$ nm) is present in each (001) planes. Such a magnetic structure agrees with the neutron-scattering measurement for bulk Cr and thick Cr films[6–8], but has not been visualized by a local probe before. Comparing with the commensurate SDW or AFM state[4,16], the IC-SDW observed here are pure spin modulations that decoupled from the lattice. It can be considered as a "modulated" ferromagnetism for the top Cr plane.

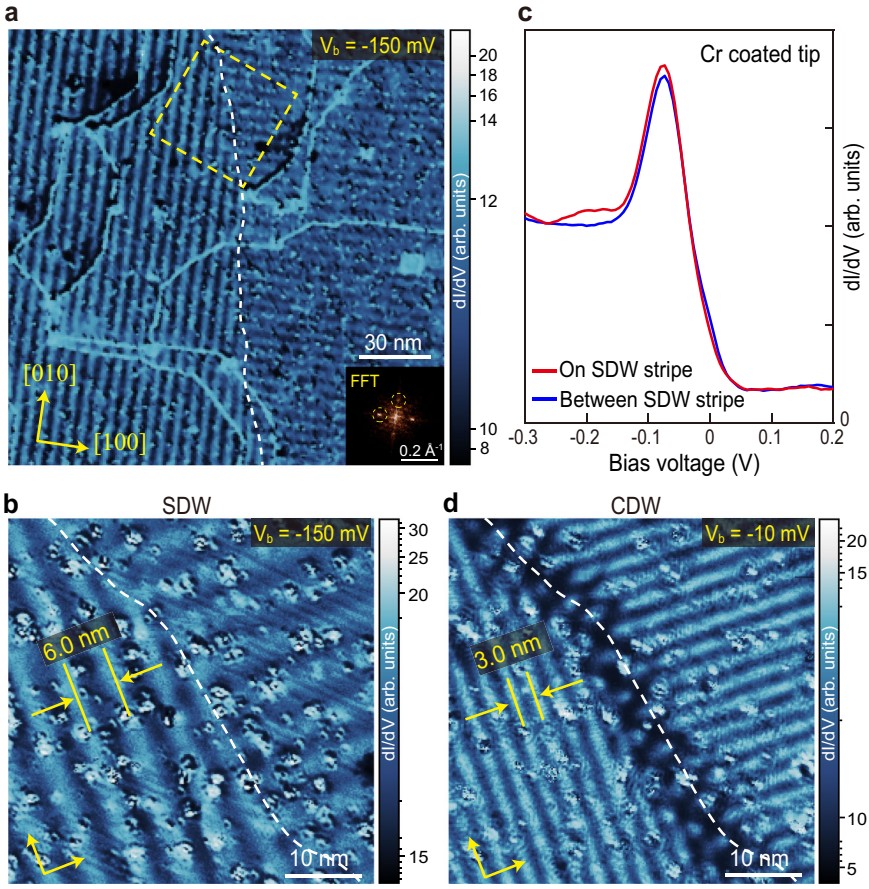

**Fig. 2 Spin and charge modulations observed on Cr (001) surface. a** The d$I$/d$V$ map of the same area that shown in Fig. 1c, taken by a Cr coated spin-polarized tip at $V_b = -150$ mV ($I = 80$ pA, $\Delta V = 20$ mV). Inset: FFT image, dashed circles indicate the spots from spin modulations. **b** A zoom-in d$I$/d$V$ map of the marked square in **a**, which shows spin modulation with a period of 6.0 nm. White dashed curve tracks the domain wall ($V_b = -150$ mV, $I = 150$ pA, $\Delta V = 15$ mV). **c** Typical d$I$/d$V$ (point spectra) taken on and in between the spin modulation stripes (setpoint: $V_b = 1$ V, $I = 60$ pA, $\Delta V = 20$ mV). **d** d$I$/d$V$ map of the same area as **b**, but taken at $V_b = -10$ mV. The charge modulation with a period of 3.0 nm is observed ($I = 150$ pA, $\Delta V = 5$ mV).

**Observation of the coexisting CDW.** Moreover, in addition to the SDW modulation, we also observed another type of modulation with half the period of SDW (3.0 nm), as shown in the d$I$/d$V$ map in Fig. 2d (taken at $V_b = -10$ mV). It displays the same domain structures with the SDW shown in Fig. 2b, however here the two domains have the same contrast. We further verified that such 3.0 nm modulation is also visible under a nonmagnetic PtIr tip, but the SDW is invisible (see Supplementary Fig. S5). Therefore, it is natural to assign such spin-unpolarized modulation to CDW with a $\mathbf{Q}_{CDW} = 2\mathbf{Q}_{SDW}$, as reported in X-ray studies of Cr[10,13]. We noticed a previous STM study on Cr (110) surface reported similar charge modulation that originated from bulk CDW[27]. Here, we are able to image the SDW and CDW simultaneously, enabling the study of their microscopic correlations.

Figure 4a–h show a series of d$I$/d$V$ maps taken in the same region, with the same Cr tip but at various $V_b$. Their (averaged) line profiles are summarized in Fig. 4j, which display the evolution of SDW/CDW modulations as the energy varies. Figure 4i shows the typical d$I$/d$V$ spectrum of this mapping region, and the energy positions corresponding to line profiles in Fig. 4j are indicated by dashed lines. The SDW modulation is mainly observable in the energy range of $-200$ meV $\sim -50$ meV, which covers the large DOS peak in d$I$/d$V$. This suggests the peak is from certain spin-polarized state(s). As the mapping energy lowered to $-50$ meV $\sim 0$, the 3.0 nm CDW modulation became pronounced. Interestingly, it displays an abrupt phase inversion between $-30$ and $-10$ meV (see also the d$I$/d$V$ maps in Fig. 4f,

g). Such π phase shift can also be seen in the phase of the Fourier transformations, as shown in Fig. 4k. We note this energy range right covers the DOS dip in the d$I$/d$V$ spectrum (Fig. 4i), which suggests such a DOS dip is from a CDW gap, as the phase inversion of particle-hole states around the gap is a hallmark of CDW[38,39]. Our measurement at elevated temperatures (shown in Supplementary Figs. S6 and S7) also suggested the DOS dip and CDW are correlated, as they are still both visible at $T = 78$ K but disappeared together at $T = 301$ K, which is close to $T_N = 311$ K.

However, it is unusual that the gap is not opened at $E_F$ here, but about 22 meV below, which is rather unexpected for conventional CDW[1]. This gap is also unlikely associated with the SDW as its size ($\approx 10$ meV) is too small with comparing to the Néel temperature of Cr (311 K). We note in previous ARPES study on Cr (110), a SDW gap is found to be about 200 meV and located above $E_F$ (ref. [14]), which can be understood through AFM coupling induced band folding (see Supplementary Fig. S8, the hole pocket of Cr is slightly larger than electron pocket, their crossing point upon folding is above $E_F$). However, here we did not observe an obvious SDW gap in tunneling spectrum, although the tunneling conductance at positive energy is indeed low in Fig. 1f (whether it is related to SDW gap needs further investigation). Assuming the Cr sample here has similar band structure to that reported in ref. [14], a CDW gap below $E_F$ cannot be induced by the band folding scenario either. It therefore suggests the formation of CDW is beyond the intuitive Fermi surface nesting picture. We note a recent STM study on TiSe$_2$ also

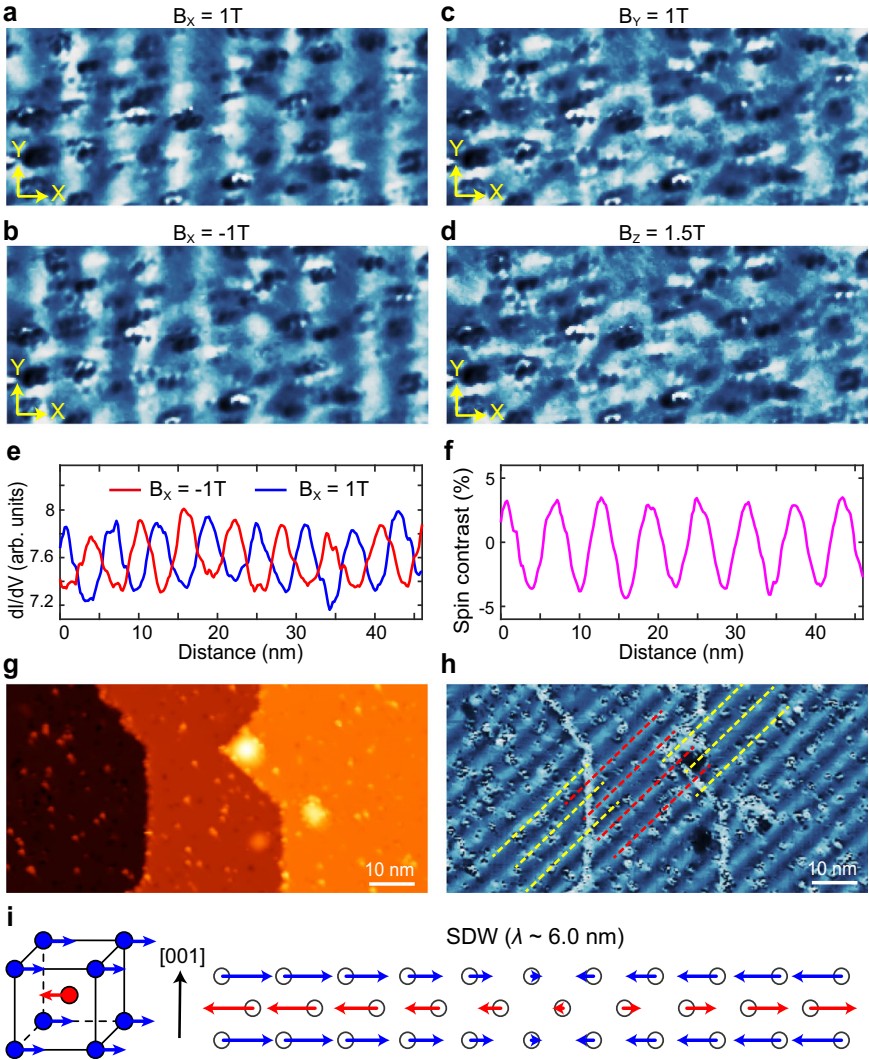

**Fig. 3 Verifying the nature of spin density wave. a–d** d$I$/d$V$ maps taken with a Fe-coated tip under different magnet field (marked above each panel). The mapping area is the same for these panels (size: 46 × 20 nm², $V_b = -100$ mV, $I = 80$ pA, $\Delta V = 10$ mV). **e** averaged line profile (along X direction) of **a**, **b**. A phase inversion can be clearly seen for $B_x = \pm 1$ T. **f** Spin contrast (polarization ratio) calculated by the relative difference of the line profiles in **e**. **g** STM image of three adjacent mono-atomic steps. The step height is ≈ 1.45 Å. **h** d$I$/d$V$ map taken in the same area of panel **g**, the SDW displays phase inversion across each step edge, as indicated by yellow and red dashed lines ($V_b = -200$ mV, $I = 100$ pA, $\Delta V = 15$ mV). **i** A sketch of local spin configuration and IC-SDW near Cr (001) surface.

indicated a CDW gap opened away from $E_F$ (ref. [40]), which was attributed to strong electron correlations.

Further information can be extracted from the real space imaging of SDW and CDW is their phase relation. As discussed before, the SDW modulations in d$I$/d$V$ are directly induced by spin contrast (Fig. 3). Their maximum and minimum positions are where the absolute spin density reach maximum. Meanwhile, the CDW modulation in d$I$/d$V$ is the local DOS variation induced by periodic charge distribution[38]. Figure 4j shows that the locations with maximum spin density (tracked by solid and dashed lines) always have minimum LDOS at −30 mV and maximum LDOS at −10 meV. As usually the charge density is proportional to the LDOS of occupied state near $E_F$, our data suggests the SDW and CDW in Cr are in-phase, i.e., the positions with maximum spin density also have maximum charge density (sketched in Fig. 1b). We note that previously such relation can only be obtained through combined X-ray and neutron diffraction measurement after extensive data analysis[41,42], while here we provide a rather direct evidence on the same system. The in-phase

relation appears to consist with the theory which treat CDW as a second harmonics of SDW[12,43,44].

At last, beside the static SDW/CDW modulations, we also observed dispersive quasi-particle interference (QPI) on the surface. As shown in Fig. 5a for example, clear short wavelength interference patterns are visible around the defects. Figure 5b is the FFT image of Fig. 5a which displays a square shaped scattering ring (more QPI data is shown in Supplementary Fig. S9). By summarizing the FFT line profile taken at various energies (Fig. 5c), an electron-like dispersion with $q_F \approx 1.1$ Å$^{-1}$ is visualized. We note previous DFT calculations had predicted multiple spin-polarized surface states on Cr(001)[33], the observed QPI could be originated from one of the surface states, as bulk bands usually do not generate strong QPI; and the DOS peak at $E = -75$ meV in d$I$/d$V$ (Fig. 1f) could be from the onset of this band. The clear observation of QPI here (which is absent in previous STM studies) is also an indication of improved surface condition. We expect it will help to elucidate the surface electronic states of Cr with the help of further theoretical calculations.

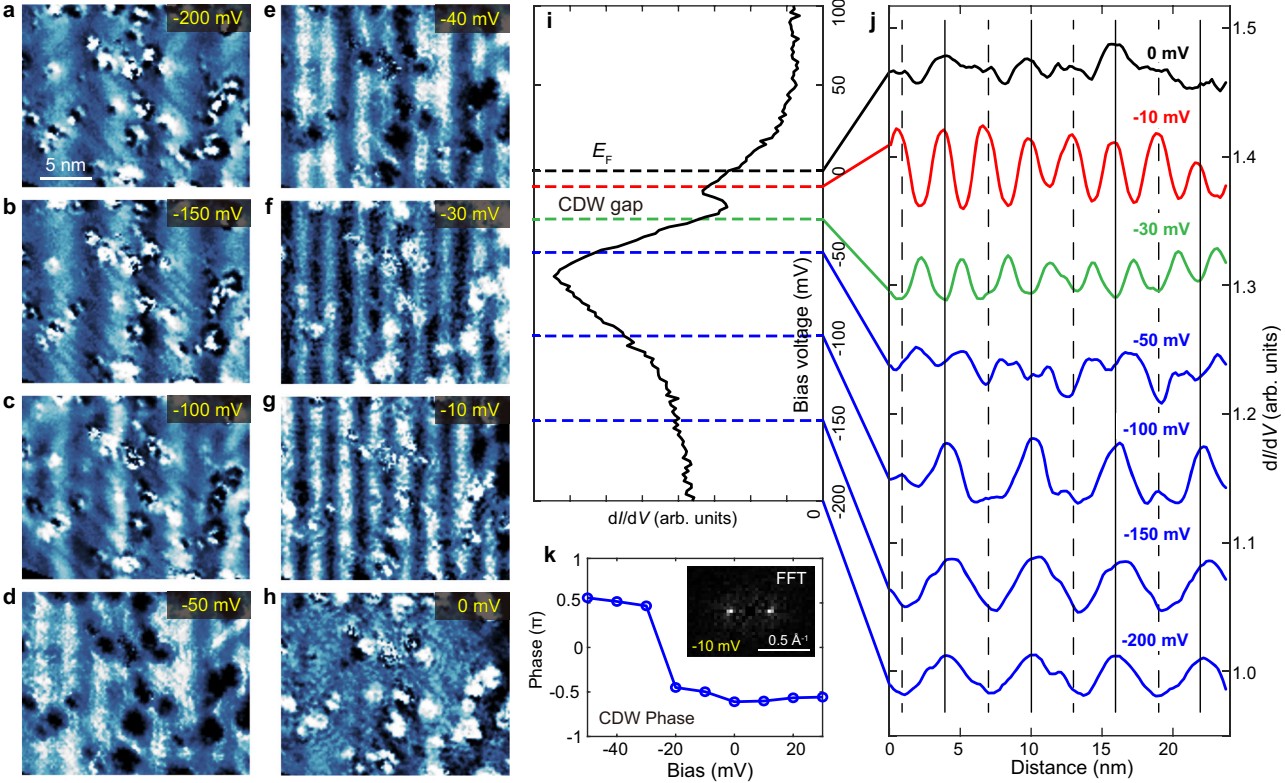

**Fig. 4 Evolvement of SDW/CDW modulations with energy, and the phase inversion of CDW. a–h** d$I$/d$V$ maps taken by Cr-coated tip at various $V_b$. The mapping area are the same for these panels. **i** The typical d$I$/d$V$ point spectrum taken in this region (setpoint: $V_b = -500$ mV, $I = 60$ pA, $\Delta V = 10$ mV). **j** Averaged line profile of the d$I$/d$V$ maps. An abrupt phase inversion can be seen between $V_b = -30$ and $-10$ mV, which corresponds to the DOS dip region in **i**. **k** The phase of the CDW modulation, which are extracted from the raw FFT values at $\mathbf{Q}_{CDW}$. Inset image shows the FFT of **g** ($V_b = -10$ mV) which displays $\mathbf{Q}_{CDW}$ spots at $\pm 2\pi/\lambda_{CDW}$.

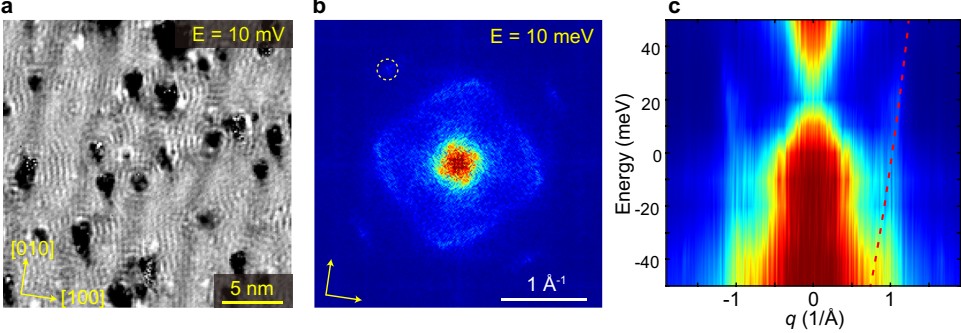

**Fig. 5 QPI measurement on Cr (001) surface. a** d$I$/d$V$ image taken at $V_b = 10$ mV ($I = 100$ pA), which shows the QPI modulation. **b** The four-fold symmetrized FFT image of at $E = 10$ meV. The dashed circle indicated the FFT spots from c($2 \times 2$) reconstruction. **c** Color plot of the summarized FFT line cuts, dashed line tracks the dispersion of electron-like band.

## Discussion

We have now presented a comprehensive SP-STM study on a well-ordered Cr (001) surface. We directly observed the incommensurate SDW with similar spin configurations to the bulk, which manifests as long-period, linearly polarized pure spin modulations on the surface; and the coexisting CDW order is also simultaneously observed. These features are absent in previous STM studies which indicates the surface condition is important for probing intrinsic magnetism of Cr. Another main finding of this work is the CDW gap opened below $E_F$. It appears conflict with conventional Fermi surface nesting picture in which the density wave gap will open at $E_F$ to lower the energy. Nonetheless, if considering the CDW in Cr is driven by SDW[12,43,44], the system would gain energy mainly through SDW and it is possible

the formation of CDW involves states away from $E_F$. This implies additional factors, such as electron correlations, may play a role in CDW of Cr. Moreover, as a real space measurement, we can directly observe the domain and phase relations of SDW and CDW. The same domain structure and an in-phase relation of local spin and charge indicate these two orders are highly correlated, consistent with the scenario that CDW is the high-order harmonics of SDW.

The mechanism of coexisting spin/charge orders has long been an important issue in condensed matter physics, particularly for correlated materials such as cuprates[3,45] and iron-based superconductors[4,46], our new spectroscopic and microscopic information provide insights on the comprehensive understanding SDW/CDW in Cr and other correlated materials. Our work is one of the

few cases in which simultaneous imaging of spin/charge order with high resolution is achieved (ref. [5] is another example). We expected similar SP-STM measurement shall also be applied to other systems and would inspire more studies on the coexisting quantum orders.

## Methods

**Cr (001) sample preparation and STM measurements.** Cr (001) single crystal (Mateck, purity: 99.999%) was intensively cleaned by repeated cycles of Ar sputtering at 750 °C (for 15 min) and annealing at 800 °C (for 20 min), until a well-ordered surface is obtained. Spin-resolved tunneling spectroscopy and conductance mapping were performed by Cr-coated and Fe-coated STM tips, which are prepared by depositing 40 nm Cr or 16 nm Fe layers on W tip. The W tip was electrochemically etched and flashed up to ≈2000 K for cleaning before coating. The tunneling conductance (d$I$/d$V$) was collected by standard lock-in method and the bias voltage ($V_b$) is applied to the sample.

## Data availability

The main data supporting the findings of this study are available within the article and its Supplementary Information files. All the raw data generated in this study are available from the corresponding author upon reasonable request.

## Code availability

All the data analysis codes related to this study are available from the corresponding author upon reasonable request.

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

## Acknowledgements

We thank Prof. Chunlei Gao and Mr. Zhongjie Wang for the advice of preparing spin-polarized tip and helpful discussions. This work is supported by the National Key R&D Program of the MOST of China with Grant Nos. 2017YFA0303004 (T.Z.), National Natural Science Foundation of China with Grant Nos. 92065202 (T.Z.), 11888101 (D.L.F.), 11790312 (D.L.F.), 11961160717 (T.Z.), Science Challenge Project with grant No. TZ2016004 (D.L.F.), Shanghai Municipal Science and Technology Major Project with Grant No. 2019SHZDZX01 (T.Z., D.L.F.), Science and Technology Commission of Shanghai Municipality, China (Grant Nos. 19JC1412702 (T.Z.), 21TQ1400100 (T.Z.)).

## Author contributions

The STM measurements and data analysis were performed by Y.N. Hu, T.Z. Zhang, D.M. Zhao, C. Chen, S.Y. Ding, W.T. Yang, X. Wang, C.H. Li, H.T. Wang and T. Zhang. D.L. Feng and T. Zhang coordinated the project and wrote the manuscript. All authors have discussed the results and the interpretation.

## Competing interests

The authors declare no competing interests.
