## [Peer Review File · Nature Communications]

Reviewers' Comments:

Reviewer #1:

Remarks to the Author:

The manuscript by Hu et al reports on a systematic study of the spin and charge density wave orders in Chromium by spin-polarized STM. The authors present a thorough study, and it is really good to see spin-polarized STM done on the classic SDW material. Their results are consistent with previous neutron scattering and x-ray studies, but provide additional information by enabling imaging of the phase relation between the XRD and neutron scattering studies.

In my opinion this manuscript should be published in Nature Communications after the following points below have been considered.

- The authors say "first time" and "new" a bit too often – I suggest removing these.
- The QPI should maybe be moved to the supplementary material, as it does not really contribute to the understanding.
- It is not clear why the CDW cannot simply be due to band hybridization. The authors argue against this scenario by comparison with ARPES measurements, however the electronic structure of Chromium is rather three-dimensional, so while it may be true that in the band structure seen by ARPES it would not lead to a hybridization gap at the energy seen in STM, is it possible that there are hybridization gaps forming in other places in the Brillouin zone?
- There are a few previous reports of real space imaging of incommensurate magnetic orders, for example in Bode et al, Nature 447, 190 and Trainer et al, Sci. Adv. 5, eaav3478. While these are spin-spirals, whereas the authors state that what they observe is a spin-density wave, it would be worth discussing the differences.
- The simultaneous observation of magnetic and charge order in spin-polarized STM has also been reported for FeTe (ref. 5).

Reviewer #2:

Remarks to the Author:

The authors image the spin density wave of Cr on its 001 surface and demonstrate that there is anti-ferromagnetic ordering along the 001 direction. The authors also observe charge density wave order. Furthermore, the authors claim that a depletion in the measured dI/dV is related to the charge density wave order and points to an unconventional form of CDW order. Finally, the authors report the observation of QPI patterns close to E_f .

The real space observation of the spin density wave order is new and provides a neat confirmation of previous results. These results are that the SDW order below the spin-flip transition is in-plane (consistent with Ref. 6 of the manuscript). The out-of-plane order is anti-ferromagnetic, which was demonstrated previously by STM measurements in Ref 19 of the manuscript. The CDW order has been previously observed (Ref. 27) and has been broadly discussed in the literature as well (for example Ref's 10 or 13). In these aspects the current manuscript brings little new insights. There are two observations that could be considered novel beyond the real space imaging of SDW order: a depletion in dI/dV and QPI patterns. Both observations could be of broader interest. The QPI patterns do not get a lot of attention in the manuscript and apart from the presentation of the result itself, it is not clear what the relevance of these are.

That leaves the claim that a depletion observed in dI/dV should be interpreted as the CDW gap and here the current manuscript does not provide a convincing case. The assignment hinges on the observation that the intensity of the CDW wave reverses between data taken at -30 meV and -10 meV.

Concerning this point I have a few questions:

- How prevalent is the depletion on the surface? Is the data presented in Fig. 1g a point spectrum or averaged over a larger area? The depletion seems to be absent in Figure 1f. Similarly, Fig. 2c does not show this depletion either. Fig. 4i is much noisier and seems to represent a point spectrum.
- Does the depletion show a temperature dependence? Does it disappear at TCDW?
- Why is the dI/dV pattern at -20 meV taken with PtIr tip in the supplementary significantly shifted

relative to the -10 and -30 meV patterns? When I look at the supplementary data in Fig. S1, there are clear CDW modulations visible at -20 meV. Why are these absent for the PtIr tip? Is there any chance of surface contamination between these different experiments?

- At -100 meV there appears to be both spin and charge density waves visible in the data presented in fig. 4j. Assuming that the large modulation is from the SDW, this leaves small peaks in the troughs of the SDW. These are phase shifted relative to the -50 meV data again. Can the authors elaborate on this?

- In previous studies (e.g. Ref. 25) the CDW modulation period was reported to vary from the bulk value of 4.2 nm to 7.7 nm for a thin film. Here a CDW period of 3 nm is reported, which seems small. Can the authors elaborate on this?

The assignment of this depletion to a CDW gap leads the authors to speculate about the nature of the CDW order. This leads to claims of 'beyond the intuitive', 'unconventional', 'provides new insight on the comprehensive understanding'. In my opinion, these claims are a bit premature at this point and I hope the authors can provide a more compelling case that the depletion in the dI/dV is a robust feature that can be conclusively tied to the CDW order.

The above does not take away that the manuscript is well organized and seems to have been written with eye for detail. As far as I can tell most necessary ingredients have been presented and discussed. There are a few small points in the manuscript that could perhaps be explained in a bit more detail or more clearly.

- Fig. 1 d: it would be nice to indicate where in 1c the image is taken.

- In Fig. 2b, 2d, 3c, 3d and 4b there is an additional modulation visible that runs in the 110 directions with a period of approximately 2nm. Why are these not mentioned and what possible interpretation could they have?

- It would be nice to know for all the presented dI/dV spectra whether they are averaged or point spectra. If they are point-spectra, it would be good to know how representative they are. One option could be to add more dI/dV spectra in a light grey color to indicate the spread.

- The color bar in Fig. 2 reads "low" and "high". Numbers are preferable.

- Fig. 3 has no color bar, but panels e and f make up for that. It would be nice if the authors could do the same for figure 4j.

- How is the phase shift obtained from the FFT? The scattering spots in the inset likely represent magnitude and therefore do not contain phase information.

- There is a sentence on page 7: "Based on the same electronic structure...". It is not clear what is meant here.

- In reviewing the manuscript I came across this reference: Spera et al. , Phys. Rev. Lett. 125, 267603 that may be relevant to the discussion presented here.

Reviewer #3:

Remarks to the Author:

Hu et al report spin-polarized scanning tunneling microscopy investigation of the surface of Cr(001). They reveal first atomic-scale signature of the spin-density wave on the surface of the material coexisting with the CDW that exhibits half the period of the SDW. The quality of STM data is high and the analysis is solid. The manuscript is also easy to follow, which I appreciate. I find the topic interesting and I can very much recommend the paper for publication with minor revisions outlined below.

(1) The authors should show the contrast reversal around -20 mV in dI/dV maps acquired with different setup conditions. It may also be useful to consider normalizing using the L-maps $(dI/dV)/(I/V)$. This normalization is often used in other STM experiments, which partially removes the effects of the setup condition. It would be beneficial, if possible, to elaborate more why they believe the CDW gap does not extend above the Fermi level. I find the work interesting either way, but I would appreciate more clarity on that.

(2) The authors should show more Fourier transforms of dI/dV maps in the Supplement, demonstrating the absence of dispersion of the Fourier peaks (SDW and CDW peaks) as a function of energy. This is a relatively standard piece of analysis used to show that the peaks are

associated with non-dispersive CDW/SDW as opposed to some dispersive QPI features. I know some FTs are showed in Fig. S3, but it is not clear what corresponds to QPI and what to CDW/SDW.

Reply to reviewers:

We thank all the reviewers for their careful review and insightful comments on our manuscript. Their comments are very helpful for improving our work. Our point-by-point responses to reviews' comments are in blue text bellow (following the original comment). The main changes in the revised manuscript are shown in highlighted text.

Reviewer #1: (Remarks to the Author):

The manuscript by Hu et al reports on a systematic study of the spin and charge density wave orders in Chromium by spin-polarized STM. The authors present a thorough study, and it is really good to see spin-polarized STM done on the classic SDW material. Their results are consistent with previous neutron scattering and x-ray studies, but provide additional information by enabling imaging of the phase relation between the XRD and neutron scattering studies. In my opinion this manuscript should be published in Nature Communications after the following points below have been considered.

We thank reviewer for pointing out the significance of our work and its potential for publication in Nature Communications.

- The authors say “first time” and “new” a bit too often – I suggest removing these.

We followed reviewer's suggestion and have removed most of these words in the revised manuscript.

- The QPI should maybe be moved to the supplementary material, as it does not really contribute to the understanding.

We agree that the QPI data appears not directly relevant to SDW/CDW and could be moved to supplementary material. Meanwhile, we also noticed that in reviewer #2's report he/she considered the QPI as new observation beyond SDW and reviewer #3 also asked about some details of QPI. Therefore, we think the QPI may still be of some interest to people (it is kind of a routine STM measurement but not reported in previous STM work). So we kept this part in the revised manuscript and we would thank reviewer for the understanding.

- It is not clear why the CDW cannot simply be due to band hybridization. The authors argue against this scenario by comparison with ARPES measurements, however the electronic structure of Chromium is rather three-dimensional, so while it may be true that in the band structure seen by ARPES it would not lead to a hybridization gap at the energy seen in STM, is it possible that there are hybridization gaps forming in other places in the Brillouin zone?

It is true that the electronic structure of Cr is very three-dimensional. The Fermi surface we draw in Fig. 1a is only a simplified 2D sketch. In Fig. R1a below we show the commonly accepted 3D Fermi surface of Cr (adopted from L. Mattheiss, Phys. Rev. 139, 1893A(1965)).

It is composed of an electron octahedron at Γ , a hole octahedron at H, some small hole ellipsoids at N points and knob-shaped electron pockets along Γ -H lines. These segments are also presented in the (001) cross section of the Fermi surface shown in Fig. R1b (adopted from D. Laurent *et al.*, PRB 23, 4977(1981)). In the ARPES studies of refs. 14 -15, neighboring AFM coupling induced band folding between electron and hole bands at Γ / H are observed, with a folding vector of $Q = 2\pi/a$ (marked in Fig. R1b). Such folding can induce a hybridization gap slightly above E_F , as sketched in Fig. R1c. However, as seen from Fig. R1a and R1b the same folding vector Q does not connect other electron and hole Fermi pockets. Therefore, we think band folding effect is unlikely to give hybridization gap in other bulk bands. One may ask if the possible surface states can participate in band hybridization. This is also not evidenced in our data since we only observed one electron-like band in QPI and no gap opening is seen (Fig. 5). We have added Fig. R1 into the revised supplementary material as Fig. S8.

Fig. R1 | (a) 3D Fermi surface of bulk Cr (L. Mattheiss, Phys. Rev. 139, 1893A(1965)). (b) Fermi surface cross section of the (001) plane (D. Laurent *et al.*, PRB 23, 4977(1981)). The folding vector between Γ and H points is $Q = 2\pi/a$. (c) A sketch showing the band folding from H to Γ which can induce a hybridization gap above E_F (the hole pocket at H is larger than the electron pocket at Γ).

- There are a few previous reports of real space imaging of incommensurate magnetic orders, for example in Bode *et al*, Nature 447, 190 and Trainer *et al*, Sci. Adv. 5, eaav3478. While these are spin-spirals, whereas the authors state that what they observe is a spin-density wave, it would be worth discussing the differences.

We thank reviewer for raising this important point. Spin-spiral is indeed a relevant spin order that need to be distinguished from SDW. As previously reported in Nature 447, 190 (2007) and Sci. Adv. 5, eaav3478 (2019), a spin-spiral consists of spins with nearly constant magnitude but whose directions rotate continuously. It is a non-collinear magnetic order usually induced by DM interactions between local spins. For a classical SDW state, the spins are linearly polarized and their magnitude displays sinusoidal modulation in real space, which is a manifestation of itinerant magnetism. In SP-STM measurement with a fixed tip magnetization, SDW and spin-spiral may both appear as stripe-like spin modulation, but they can be distinguished by changing the magnetization of the tip (e.g., using vector magnetic field). In our measurement, the disappearance of modulation under B field along Y and Z direction means

the spin is only polarized along X (Fig. 3a-d), which suggested a SDW state. In Bode et al. and Trainer et al.'s papers, the spin-spiral states are verified by the observation of (phase shifted) modulations under different tip magnetizations. We have added discussions about spin-spirals in revised manuscript (page 6, 3rd paragraph, highlighted) and cited the two papers mentioned above as refs. 36 and 37.

- The simultaneous observation of magnetic and charge order in spin-polarized STM has also been reported for FeTe (ref. 5).

We thank reviewer for reminding that ref. 5 has also reported simultaneous imaging of (commensurate) CDW and AFM orders. We have added corresponding statement in the main text (see discussion part, page 11, highlighted text).

Reviewer #2: (Remarks to the Author):

The authors image the spin density wave of Cr on its 001 surface and demonstrate that there is anti-ferromagnetic ordering along the 001 direction. The authors also observe charge density wave order. Furthermore, the authors claim that a depletion in the measured dI/dV is related to the charge density wave order and points to an unconventional form of CDW order. Finally, the authors report the observation of QPI patterns close to E_f .

The real space observation of the spin density wave order is new and provides a neat confirmation of previous results. These results are that the SDW order below the spin-flip transition is in-plane (consistent with Ref. 6 of the manuscript). The out-of-plane order is anti-ferromagnetic, which was demonstrated previously by STM measurements in Ref 19 of the manuscript. The CDW order has been previously observed (Ref. 27) and has been broadly discussed in the literature as well (for example Ref's 10 or 13). In these aspects the current manuscript brings little new insights.

There are two observations that could be considered novel beyond the real space imaging of SDW order: a depletion in dI/dV and QPI patterns. Both observations could be of broader interest. The QPI patterns do not get a lot of attention in the manuscript and apart from the presentation of the result itself, it is not clear what the relevance of these are.

We thank reviewer for the careful reading of our manuscript. Reviewer also pointed out our new findings besides the real-space imaging of SDW, which are the dip feature in dI/dV and the QPI pattern. The QPI measurement was aimed at gathering information about surface condition and electronic structures, which is a kind of routine measurement in STM study, but we agree that its direct relation to SDW/CDW is unclear at this stage.

That leaves the claim that a depletion observed in dI/dV should be interpreted as the CDW gap and here the current manuscript does not provide a convincing case. The assignment hinges on the observation that the intensity of the CDW wave reverses between data taken at -30 meV and -10 meV.

Concerning this point I have a few questions:

- How prevalent is the depletion on the surface? Is the data presented in Fig. 1g a point spectrum or averaged over a larger area? The depletion seems to be absent in Figure 1f. Similarly, Fig. 2c does not show this depletion either. Fig. 4i is much noisier and seems to represent a point spectrum.

The DOS dip near $-22(\pm 1)$ mV is a common feature on the surface. We now show more dI/dV spectra in Fig. S2 of revised supplementary material and attached it below as Fig. R2. Fig. R2b and R2c are two series of spectra taken along the two lines marked in Fig. R2a (measured by PtIr tip). All of them display a similar dip around $-22(\pm 1)$ mV. We also repeatedly observed the dip in dI/dV spectra taken with different tips. The data shown in Fig. 1g is a point spectrum taken by a W tip, and Fig. 4i is a point spectrum taken by Cr coated tip. The dip is absent in Fig. 1f because its energy scale ($\pm 1V$) is too large for resolving this small dip (about 20 mV wide). The spectra in Fig. 2c are taken by relatively large Bias voltage modulation ($\Delta V = 20$ meV) so the dip is not obvious, but a kink-like feature can be seen near -20 meV. In Fig. R3 below we show additional spectra taken within smaller energy scale or smaller Bias modulation (measured by either a W or Cr-coated tip), the DOS dip can be seen in all of them.

Fig. R2 | Spatial dependence of the dI/dV spectra (Pt/Ir tip, $T = 5.0K$). (a) Topographic image of Cr(001) ($V_b = -30$ mV, $I = 10$ pA). (b - c) Series of dI/dV spectra taken along the two arrows shown in panel a, respectively (setpoint: $V_b = -100$ mV, $I = 150$ pA, $\Delta V = 5$ mV). The DOS dip at $E = -22(\pm 1)$ meV appears in every spectrum. The spectra are shifted vertically. (d) A non-shifted plot of all the spectra that shown in panel b and c, the red curve is averaged spectrum.

Fig. R3 | (a) dI/dV spectra taken by W tip within $\pm 0.5V$ (setpoint: $V_b = -500$ mV, $I = 80$ pA, $\Delta V = 10$ mV), a notable kink near -22 meV is seen. (b) dI/dV spectra taken by W tip in a smaller energy window (setpoint: $V_b = -300$ mV, $I = 60$ pA, $\Delta V = 5$ mV), the dip near -22 meV is clearly seen. (c) dI/dV spectra taken by Cr coated tip (setpoint: $V_b = -300$ mV, $I = 60$ pA, $\Delta V = 5$ mV). The dip near -22 meV is also seen.

- Does the depletion show a temperature dependence? Does it disappear at TCDW?

This is a good point raised by reviewer. The Néel temperature (T_N) of Cr is 311 K and T_{CDW} was reported to be the same with T_N (ref. 6). To check the temperature dependence, we have performed STM measurements at LN₂ temperature (78K) and room temperature (301K) (It is difficult to continuously vary the temperature of our cryogenic STM system). The results are shown in Fig. S6 and Fig. S7 of the revised supplementary material and attached below as Fig. R4 and Fig. R5.

At $T = 78K$, the spatially averaged dI/dV spectrum shows a kink like feature around -23 mV (Fig. R4b). CDW modulations are still present in dI/dV maps (Fig. R4c-h) and the phase inversion between -30 and -10 mV is also observed (Fig. R4i). The kink feature in dI/dV indicates the DOS depletion is still present albeit it is significantly broadened (the thermal broadening (~ 3.5 k_BT) at 78K is about 23 meV). This is consistent with our assumption that the DOS depletion is from CDW gap, which should exist at temperatures far below T_N .

At $T = 301K$ (which is $\approx 97\%$ of T_N), CDW modulations are not visible in dI/dV maps (Fig. R5d-f), and the dI/dV spectrum is featureless in ± 50 meV range (Fig. R5c). Nevertheless, the DOS peak at -75 meV is still visible in a larger scale spectrum albeit much broadened (Fig. R5b). These results are consistent with that the CDW/SDW should disappear near T_N , while the basic band structure of Cr will not change.

Fig. R4. Measurement at $T = 78$ K by Pt/Ir tip. (a) Topographic image of the Cr(001) surface ($V_b = -40$ mV, $I = 50$ pA). (b) Spatially averaged dI/dV spectrum (setpoint: $I = 100$ pA, $V_b = -100$ mV, $\Delta V = 1$ mV). A kink feature is observed around -23 meV. (c - h) dI/dV maps taken in the same area of panel a and at various energies (setpoint: $I = 50$ pA, $V_b = -50$ mV, $\Delta V = 8$ mV). (i) Line profiles of the dI/dV maps (averaged within the region marked in panel c). A phase inversion can be seen between $V_b = -30$ mV and -20 mV.

Fig. R5 | Measurement at T = 301K by PtIr tip. (a) Topographic image of Cr(001) surface ($V_b = -30$ mV, $I = 100$ pA). (b - c) dI/dV spectra measured at defect free area. (setpoint: b, $I = 100$ pA, $V_b = -1.0$ V, $\Delta V = 10$ mV; c, $I = 100$ pA, $V_b = -0.2$ V, $\Delta V = 1$ mV). (d - f) Several dI/dV maps taken in the same area as panel a (setpoint: $I = 100$ pA; V_b is labeled in each panel; $\Delta V = 3$ mV). No CDW modulation is observed.

- Why is the dI/dV pattern at -20 meV taken with PtIr tip in the supplementary significantly shifted relative to the -10 and -30 meV patterns? When I look at the supplementary data in Fig. S1, there are clear CDW modulations visible at -20 meV. Why are these absent for the PtIr tip?

The dI/dV maps taken by PtIr tip (now shown in Fig. S5) are all measured in the same area of Fig. S5a. Reviewer may mean that the background dI/dV pattern seems to be different in different dI/dV maps. We think these non-uniform background DOS could be induced by different types of surface defects that seen Fig. S5a. Nevertheless, we found there are still common features in DOS maps induced by certain defects. We have indicated some of these features by arrows in Fig. S5.

For the -20 meV map measured by Cr coated tip (now shown in Fig. S3h), it indeed displays modulation but has weaker intensity than that in -30 meV and -10 meV maps (Fig. S3g, 3i). And it appears to have both 3.0 nm and 6.0 nm period. Thus we think the modulation should be partially contributed by SDW. We also noticed the bias voltage modulation (ΔV) used for Fig. S3h is 5 meV, while the ΔV used for the -20 meV map taken by PtIr tip in Fig. S5 is 3 meV. Since large bias modulation can broaden the mapping energy window, this may explain why the -20 meV map in Fig. S3h shows some CDW modulation (-20 meV is not exactly the center of the DOS dip). In the measurement performed at $T = 78$ K shown in Fig. R4 above, weak CDW modulation was also observed in -20 meV maps, which could be due to the thermal broadening of the CDW gap. Nevertheless, a phase inversion of CDW is still observed between -30 meV and -20 meV maps, and the DOS dip is evidenced around -23 meV. Therefore, we think the presence/absence of weak modulations near DOS dip should not affect our main conclusion. The key observation is still the phase inversion when the energy crossed the DOS dip.

- At -100 meV there appears to be both spin and charge density waves visible in the data presented in fig. 4j. Assuming that the large modulation is from the SDW, this leaves small peaks in the troughs of the SDW. These are phase shifted relative to the -50 meV data again. Can the authors elaborate on this?

Fig. 4j is the averaged line profile of Fig. 4c (averaged vertically). In Fig. 4c there are indeed some features in between SDW stripes but they are kind of irregular and hard to be distinguished from defect states. We also measured dI/dV map at -100 meV with different tips, as the ones shown in Fig. 3a,b (Fe coated tip) and Fig. S5e (PtIr tip). However, none of these maps show evidence of CDW. Therefore we think the small feature in troughs of SDW in Fig. 4j are likely from occasionally emerged defect states.

- In previous studies (e.g. Ref. 25) the CDW modulation period was reported to vary from the bulk value of 4.2 nm to 7.7 nm for a thin film. Here a CDW period of 3 nm is reported, which seems small. Can the authors elaborate on this?

The CDW in bulk Cr has a period of 3.0 nm at $T < 10\text{K}$ (ref. 6,10), and propagates along one of the $\langle 001 \rangle$ directions. The value of 4.2 nm mentioned by reviewer should be measured by STM on Cr(110) surface (ref. 25, 27), as (110) surface projected CDW (propagating along $[100]/[010]$) has a period of $\sqrt{2} \times 3.0 \text{ nm} \approx 4.2 \text{ nm}$ (illustrated in figure 3 of ref. 27). This is consistent with our measurement on a (001) surface in which in-plane propagating CDW should display the same period to the bulk value. In thin Cr films ($< 12 \text{ nm}$), a gradual increase of CDW/SDW period was observed as the film thickness decreases (ref. 25 and 27). It was explained by the reorientation of the Q_{SDW} , possibly due to the surface/interface contribution to the total energy. For thick films, a CDW/SDW period same to the bulk value is still observed (ref. 27). Here since we measured a Cr single crystal, we expect to see a bulk value of CDW/SDW period.

The assignment of this depletion to a CDW gap leads the authors to speculate about the nature of the CDW order. This leads to claims of ‘beyond the intuitive’, ‘unconventional’, ‘provides new insight on the comprehensive understanding’. In my opinion, these claims are a bit premature at this point and I hope the authors can provide a more compelling case that the depletion in the dI/dV is a robust feature that can be conclusively tied to the CDW order.

We agree with reviewer that we should be very careful to assign the DOS dip to CDW gap. During this revision we performed additional STM measurement at elevated temperatures (78K and 301K). The results suggest that the DOS dip and CDW modulation persist at 78K ($< T_N$), and both disappear as T approaches T_N (311K). We also show that the DOS dip around -22 meV is a robust feature in the spectra of different surface locations and in the spectra taken by different tips. The CDW always display a π phase shift around the DOS dip. Therefore, the DOS dip is very likely tied to the CDW order.

The above does not take away that the manuscript is well organized and seems to have been written with eye for detail. As far as I can tell most necessary ingredients have been presented and discussed. There are a few small points in the manuscript that could perhaps be explained in a bit more detail or more clearly.

- Fig. 1 d: it would be nice to indicate where in 1c the image is taken.

We have indicated the region where Fig. 1d was taken in Fig. 1c by a dashed square.

- In Fig. 2b, 2d, 3c, 3d and 4b there is an additional modulation visible that runs in the 110 directions with a period of approximately 2nm. Why are these not mentioned and what possible interpretation could they have?

Reviewer is very observant in pointing out this detail. These small stripe-like features are

from local dislocation lines on the surface, which can also be seen in topographic image as the one shown in Fig. 1d and Fig. R6a below (black dashed lines). It can be seen that the lattice on one side of the dislocation line gradually shifts (starting from the two arrows) with respect to the lattice on the other side, as indicated by the green/blue solid lines. The exact origin of these dislocation lines is not clear but they are likely from the underneath Cr lattice (the dislocation gradually emerge, starting from the end of the lines). In the dI/dV maps the dislocation lines appear as short stripe-like feature along $[1\bar{1}0]$, but their distribution is rather random, as seen in Fig. R6b below and Figs. 2,3,4. They do not show influence to SDW/CDW modulation so we didn't discuss them in previous manuscript. In the revised manuscript we have pointed out this feature in main text (page 4, 1st paragraph, highlighted) and added related descriptions in the supplementary material (Fig. S1 and its caption).

Fig. R6. The surface dislocation lines and corresponding dI/dV map. (a) Topographic image ($8 \times 8 \text{ nm}^2$) taken at the dashed box area in panel b. The dislocation lines are indicated by dashed line. The lattice shift around them are indicated by blue/green solid lines. (b) dI/dV map taken by a Cr coated tip ($V_b = -150 \text{ mV}$, $I = 80 \text{ pA}$, $\Delta V = 20 \text{ mV}$, $60 \times 60 \text{ nm}^2$). The short stripes along the $1\bar{1}0$ directions are caused by the dislocation lines as that shown in panel a.

- It would be nice to know for all the presented dI/dV spectra whether they are averaged or point spectra. If they are point-spectra, it would be good to know how representative they are. One option could be to add more dI/dV spectra in a light grey color to indicate the spread.

We have specified whether the dI/dV spectra are point spectra or averaged spectra in all the figure captions. We now show more spectra in Fig. S2. Fig. S2d plots overlapped spectra with light grey color, which indicates their spread. It is seen that both the -75 meV peak and the -22 meV dip are robust features.

- The color bar in Fig. 2 reads “low” and “high”. Numbers are preferable.

We have added values and ticks to the color bar of Figs. 2a,2b,2d. Please note we used a nonlinear color map to enhance the contrast of SDW/CDW modulations and suppress the

contrast from defects (Gwyddion software was used to process the images). In nonlinear color map a data value range which contains more data points will occupy a wider color range, as indicated by the ticks of the color bar.

- Fig. 3 has no color bar, but panels e and f make up for that. It would be nice if the authors could do the same for figure 4j.

We have added dI/dV values for Figure 4j on the right side. All the curves in Figure 4j are normalized by their mean value, and shifted vertically by an interval of 0.08 (except the -200 mV curve at the bottom). The relative strength of SDW/CDW modulations can still be read out from the figure.

- How is the phase shift obtained from the FFT? The scattering spots in the inset likely represent magnitude and therefore do not contain phase information.

The CDW phase plotted in Fig. 4k is directly extracted from the raw FFT values at the Q_{CDW} spots (the phase values only have relative meaning); while the inset image only shows FFT magnitude. We have added description on how to obtain the phase (in Fig. 4 caption).

- There is a sentence on page 7: “Based on the same electronic structure...”. It is not clear what is meant here.

Here we mean that if our Cr sample has similar electronic structure to that reported by ARPES in ref. 14, the band folding effect cannot induce a gap below E_F (which is now more clearly illustrated in Fig. S8). We have revised this sentence to make the meaning clear.

- In reviewing the manuscript I came across this reference: Spera et al., Phys. Rev. Lett. 125, 267603 that may be relevant to the discussion presented here.

We thank reviewer for letting us know this important reference. In this paper Spera et al. had observed a contrast inversion in STM image which indicates a CDW gap opening away from E_F . We have cited this paper in the revised manuscript (as ref. 40) and add some related statement (page 8, 3rd paragraph, last sentence).

Reviewer #3 (Remarks to the Author):

Hu et al report spin-polarized scanning tunneling microscopy investigation of the surface of Cr(001). They reveal first atomic-scale signature of the spin-density wave on the surface of the material coexisting with the CDW that exhibits half the period of the SDW. The quality of STM data is high and the analysis is solid. The manuscript is also easy to follow, which I appreciate. I find the topic interesting and I can very much recommend the paper for publication with minor revisions outlined below.

We thank reviewer for his/her interest on our work and strong recommendation of publication in Nature Communications.

(1) The authors should show the contrast reversal around -20 mV in dI/dV maps acquired with different setup conditions. It may also be useful to consider normalizing using the L-maps $(dI/dV)/(I/V)$. This normalization is often used in other STM experiments, which partially removes the effects of the setup condition. It would be beneficial, if possible, to elaborate more why they believe the CDW gap does not extend above the Fermi level. I find the work interesting either way, but I would appreciate more clarity on that.

We understand that the setup condition may affect the contrast of dI/dV maps. We have performed dI/dV mapping of CDW at different setup conditions. The maps in Figs. 4d - 4h were taken at the same set point of $V_b = -50$ mV, $I = 150$ pA, in which a contrast reversal is observed between -30 mV and -10 mV. Meanwhile, Figs. S5d, S5b of revised supplementary material show -30 mV and -10 mV maps taken at different set points: the former is $V_b = -30$ mV, $I = 100$ pA and the latter is $V_b = -10$ mV, $I = 100$ pA. A contrast reversal is also observed between them. Figs. S5b, S5d are equivalent with their L-maps $(dI/dV)/(I/V)$ that mentioned by reviewer, as all the data points in each of these maps are measured at the same I and same V . We thank reviewer for pointing out the importance of setup condition. In the revised manuscript we have specified the setup condition of all the dI/dV maps in figure captions.

We consider the CDW gap sitting below E_F is just based on the observation that the DOS dip and CDW phase inversion both happen around -22 mV. The full width of this dip appears to be small (≈ 20 meV) which does not extend above E_F . Meanwhile, no other phase change is observed above E_F (Fig. 4k) and actually the intensity of CDW modulation is much weakened at positive energies (see Fig. S3 and the FFT in Fig. S4).

(2) The authors should show more Fourier transforms of dI/dV maps in the Supplement, demonstrating the absence of dispersion of the Fourier peaks (SDW and CDW peaks) as a function of energy. This is a relatively standard piece of analysis used to show that the peaks are associated with non-dispersive CDW/SDW as opposed to some dispersive QPI features. I know some FTs are showed in Fig. S3, but it is not clear what corresponds to QPI and what to CDW/SDW.

We now show another set of FFT images in Fig. S4 of revised supplementary materials and attached it below as Fig. R7. They are the raw Fourier transforms of the dI/dV map in Fig. S3. Since there are two SDW/CDW domains in the real-space map, the FTs show two sets of scattering spots corresponding to SDW/CDW arranged in perpendicular directions. Figs. S4o, S4p summarized the FFT line-profiles taken across SDW/CDW spots of these two domains (marked in Fig. S4a). It is seen that the $Q_{SDW/CDW}$ peaks barely shift with energy, as tracked by dashed lines sitting at $Q_{CDW} = 0.21 \text{ \AA}^{-1}$ and $Q_{SDW} = 0.105 \text{ \AA}^{-1}$.

For QPI, its modulation period ($\sim 0.55 - 0.65$ nm) is significantly smaller than the SDW/CDW and the maps in Fig. S3 do not have enough resolution to reveal them. We then measured the QPI in a smaller area shown in Fig. 5a and Fig. S9. In these maps, although the SDW/CDW modulation can still be seen, their q -space spots are overwhelmed by the noisy

background at the center of FFT images (see Fig. S9 and Fig. 5c, the low- q noise is from randomly distributed defects). Therefore we can only extract the QPI dispersion in Fig. 5c.

Fig. R7 | FFT images and the non-dispersive behavior of SDW/CDW. (a - n) Raw FFT images of the dI/dV maps shown in Fig. S3(a - n), respectively. Two domains of the SDW/CDW perpendicular to each other give rise to two sets of SDW/CDW spots in FFT. The red and green dashed circles in panels **b** and **f** indicate the SDW and CDW spots, respectively. (o - p) FFT line-profiles at different energies, taken along the two yellow dashed lines marked in panel **a** (respectively). The green and red dashed lines indicate the positions of $Q_{CDW} = 0.21 \text{ \AA}^{-1}$ and $Q_{SDW} = 0.105 \text{ \AA}^{-1}$, respectively.

Reviewers' Comments:

Reviewer #1:

Remarks to the Author:

The authors have accounted for most of my suggestions. They have left the QPI data in the main manuscript, but I don't have a particular objection to that. I am happy for the manuscript to be published as is.

Reviewer #2:

Remarks to the Author:

I would like to thank the authors for their extensive replies. They have clarified the questions I had and I can at this point only recommend the paper to be published.

Reviewer #3:

Remarks to the Author:

I read the response to my comments, and those by other Reviewers. I think the authors sufficiently addressed all questions/concerns, and I (still) recommend the publication in Nature Communications.

Reply to reviewers:

We thank all the reviewers for their time and careful review on our manuscript very much. Below are our responses to reviewers' comments.

Reviewer #1 (Remarks to the Author):

The authors have accounted for most of my suggestions. They have left the QPI data in the main manuscript, but I don't have a particular objection to that. I am happy for the manuscript to be published as is.

We thank reviewer's consideration and the recommendation of publication.

Reviewer #2 (Remarks to the Author):

I would like to thank the authors for their extensive replies. They have clarified the questions I had and I can at this point only recommend the paper to be published.

We thank reviewer's recommendation of publication.

Reviewer #3 (Remarks to the Author):

I read the response to my comments, and those by other Reviewers. I think the authors sufficiently addressed all questions/concerns, and I (still) recommend the publication in Nature Communications.

We thank reviewer's recommendation of publication.